# When in Doubt, SWAP: High-Dimensional Sparse Recovery from Correlated Measurements

**Divyanshu Vats**
Rice University
Houston, TX 77251
dvats@rice.edu

**Richard Baraniuk**
Rice University
Houston, TX 77251
richb@rice.edu

## Abstract

We consider the problem of accurately estimating a high-dimensional sparse vector using a small number of linear measurements that are contaminated by noise. It is well known that standard computationally tractable sparse recovery algorithms, such as the Lasso, OMP, and their various extensions, perform poorly when the measurement matrix contains highly correlated columns. We develop a simple greedy algorithm, called SWAP, that iteratively *swaps* variables until a desired loss function cannot be decreased any further. SWAP is surprisingly effective in handling measurement matrices with high correlations. We prove that SWAP can easily be used as a wrapper around standard sparse recovery algorithms for improved performance. We theoretically quantify the statistical guarantees of SWAP and complement our analysis with numerical results on synthetic and real data.

## 1 Introduction

An important problem that arises in many applications is that of recovering a high-dimensional sparse (or approximately sparse) vector given a small number of linear measurements. Depending on the problem of interest, the unknown sparse vector can encode relationships between genes [1], power line failures in massive power grid networks [2], sparse representations of signals [3, 4], or edges in a graphical model [5,6], to name just a few applications. The simplest, but still very useful, setting is when the observations can be approximated as a *sparse* linear combination of the columns in a measurement matrix $X$ weighted by the non-zero entries of the unknown sparse vector. In this paper, we study the problem of recovering the location of the non-zero entries, say $S^*$, in the unknown vector, which is equivalent to recovering the columns of $X$ that $y$ depends on. In the literature, this problem is often to referred to as the *sparse recovery* or the *support recovery* problem.

Although several tractable sparse recovery algorithms have been proposed in the literature, statistical guarantees for accurately estimating $S^*$ can only be provided under conditions that limit how correlated the columns of $X$ can be. For example, if there exists a column, say $X_i$, that is nearly linearly dependent on the columns indexed by $S^*$, some sparse recovery algorithms may falsely select $X_i$. In certain applications, where $X$ can be specified a priori, correlations can easily be avoided by appropriately choosing $X$. However, in many applications, $X$ cannot be specified by a practitioner, and correlated measurement matrices are inevitable. For example, when the columns in $X$ correspond to gene expression values, it has been observed that genes in the same pathway produce correlated values [1]. Additionally, it has been observed that regions in the brain that are in close proximity produce correlated signals as measured using an MRI [7].

In this paper, we develop new sparse recovery algorithms that can accurately recover $S^*$ for measurement matrices that exhibit strong correlations. We propose a greedy algorithm, called SWAP, that iteratively *swaps* variables starting from an initial estimate of $S^*$ until a desired loss function cannot be decreased any further. We prove that SWAP can accurately identify the true signal support

under relatively mild conditions on the restricted eigenvalues of the matrix $X^T X$ and under certain conditions on the correlations between the columns of $X$. A novel aspect of our theory is that the conditions we derive are only needed when conventional sparse recovery algorithms fail to recover $S^*$. This motivates the use of SWAP as a wrapper around sparse recovery algorithms for improved performance. Finally, using numerical simulations, we show that SWAP consistently outperforms many state of the art algorithms on both synthetic and real data corresponding to gene expression values.

As alluded to earlier, several algorithms now exist in the literature for accurately estimating $S^*$. The theoretical properties of such algorithms either depend on the irrepresentability condition [5, 8–10] or various forms of the restricted eigenvalue conditions [11,12]. See [13] for a comprehensive review of such algorithms and the related conditions. SWAP is a greedy algorithm with novel guarantees for sparse recovery and we make appropriate comparisons in the text. Another line of research when dealing with correlated measurements is to estimate a superset of $S^*$; see [14–18] for examples.

The rest of the paper is organized as follows. Section 2 formally defines the sparse recovery problem. Section 3 introduces SWAP. Section 4 presents theoretical results on the conditions needed for provably correct sparse recovery. Section 5 discusses numerical simulations. Section 6 summarizes the paper and discusses future work.

## 2   Problem Setup

Throughout this paper, we assume that $y \in \mathbb{R}^n$ and $X \in \mathbb{R}^{n \times p}$ are known and related to each other by the linear model

$$y = X\beta^* + w\,, \tag{1}$$

where $\beta^* \in \mathbb{R}^p$ is the unknown sparse vector that we seek to estimate. We assume that the columns of $X$ are normalized, i.e., $\|X_i\|_2^2 / n = 1$ for all $i \in [p]$, where we use the notation $[p] = \{1, 2, \ldots, p\}$ throughout the paper. In practice, normalization can easily be done by scaling $X$ and $\beta^*$ accordingly. We assume that the entries of $w$ are i.i.d. zero-mean sub-Gaussian random variables with parameter $\sigma$ so that $\mathbb{E}[\exp(tw_i)] \leq \exp(t^2\sigma^2/2)$. The sub-Gaussian condition on $w$ is common in the literature and allows for a wide class of noise models, including Gaussian, symmetric Bernoulli, and bounded random variables. We let $k$ be the number of non-zero entries in $\beta^*$, and let $S^*$ denote the location of the non-zero entries. It is common to refer to $S^*$ as the support of $\beta^*$ and we adopt this notation throughout the paper.

Once $S^*$ has been estimated, it is relatively straightforward to estimate $\beta^*$. Thus, we mainly focus on the sparse recovery problem of estimating $S^*$. A classical strategy for sparse recovery is to search for a support of size $k$ that minimizes a suitable loss function. For a support $S$, we assume the least-squares loss, which is defined as follows:

$$\mathcal{L}(S; y, X) := \min_{\alpha \in \mathbb{R}^{|S|}} \|y - X_S\alpha\|_2^2 = \left\|\Pi^\perp[S]y\right\|_2^2\,, \tag{2}$$

where $X_S$ refers to an $n \times |S|$ matrix that only includes the columns indexed by $S$ and $\Pi^\perp[S] = I - X_S(X_S^T X_S)^{-1} X_S^T$ is the orthogonal projection onto the null space of the linear operator $X_S$. In this paper, we design a sparse recovery algorithm that provably, and efficiently, finds the true support for a broad class of measurement matrices that includes matrices with high correlations.

## 3   Overview of SWAP

We now describe our proposed greedy algorithm SWAP. Recall that our main goal is to find a support $\widehat{S}$ that minimizes the loss defined in (2). Suppose that we are given an estimate, say $S^{(1)}$, of the true support and let $L^{(1)}$ be the corresponding least-squares loss (see (2)). We want to transition to another estimate $S^{(2)}$ that is closer (in terms of the number of true variables), or equal, to $S^*$. Our main idea to transition from $S^{(1)}$ to an appropriate $S^{(2)}$ is to *swap* variables as follows:

*Swap every $i \in S^{(1)}$ with $i' \in (S^{(1)})^c$ and compute the resulting loss $L_{i,i'}^{(1)} = \mathcal{L}(\{S^{(1)}\backslash i\} \cup i'; y, X)$.*

If $\min_{i,i'} L_{i,i'}^{(1)} < L^{(1)}$, there exists a support that has a lower loss than the original one. Subsequently, we find $\{\widehat{i}, \widehat{i'}\} = \arg\min_{i,i'} L_{i,i'}^{(1)}$ and let $S^{(2)} = \{S^{(1)}\backslash\widehat{i}\} \cup \{\widehat{i'}\}$. We repeat the

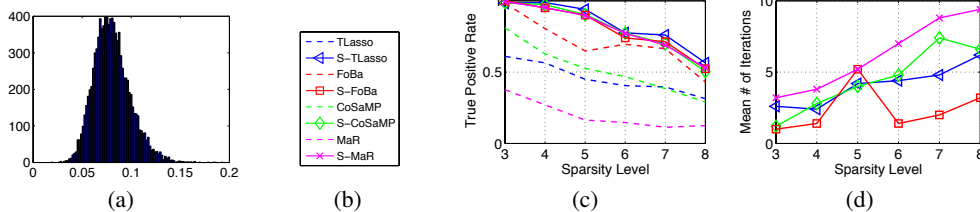

Figure 1: Example of using SWAP on pseudo real data where the design matrix $X$ corresponds to gene expression values and $y$ is simulated. The notation S-Alg refers to the SWAP based algorithms. (a) Histogram of sparse eigenvalues of $X$ over $10,000$ random sets of size 10; (b) legend; (c) mean true positive rate vs. sparsity; (d) mean number of iterations vs. sparsity.

---

**Algorithm 1:** SWAP$(y, X, S)$

---

*Inputs:* Measurements $y$, design matrix $X$, and initial support $S$.

1  Let $r = 1$, $S^{(1)} = S$, and $L^{(1)} = \mathcal{L}(S^{(1)}; y, X)$

2  Swap $i \in S^{(r)}$ with $i' \in (S^{(r)})^c$ and compute the loss $L_{i,i'}^{(r)} = L(\{S^{(r)} \backslash i\} \cup i'; y, X)$.

3  **if** $\min_{i,i'} \mathcal{L}_{i,i'}^{(r)} < L^{(r)}$ **then**

4  $\quad$ $\{\widehat{i}, \widehat{i'}\} = \text{argmin}_{i,i'} \mathcal{L}_{i,i'}^{(r)}$ (In case of a tie, choose a pair arbitrarily)

5  $\quad$ Let $S^{(r+1)} = \{S^{(r)} \backslash \widehat{i}\} \cup \widehat{i'}$ and $L^{(r+1)}$ be the corresponding loss.

6  $\quad$ Let $r = r + 1$ and repeat steps 2-4.

$\quad$ **else**

7  $\quad$ Return $\widehat{S} = S^{(r)}$.

---

above steps to find a sequence of supports $S^{(1)}, S^{(2)}, \ldots, S^{(r)}$, where $S^{(r)}$ has the property that $\min_{i,i'} L_{i,i'}^{(r)} \geq L^{(r)}$. In other words, we stop SWAP when perturbing $S^{(r)}$ by one variable increases or does not change the resulting loss. These steps are summarized in Algorithm 1.

Figure 1 illustrates the performance of SWAP for a matrix $X$ that corresponds to 83 samples of 2308 gene expression values for patients with small round blue cell tumors [19]. Since there is no ground truth available, we simulate the observations $y$ using Gaussian $w$ with $\sigma = 0.5$ and randomly chosen sparse vectors with non-zero entries between 1 and 2. Figure 1(a) shows the histogram of the eigenvalues of 10,000 randomly chosen matrices $X_A^T X_A / n$, where $|A| = 10$. We clearly see that these eigenvalues are very small. This means that the columns of $X$ are highly correlated with each other. Figure 1(c) shows the mean fraction of variables estimated to be in the true support over 100 different trials. Figure 1(d) shows the mean number of iterations required for SWAP to converge.

*Remark* 3.1. The main input to SWAP is the initial support $S$. This parameter implicitly specifies the desired sparsity level. Although SWAP can be used with a random initialization $S$, we recommend using SWAP in combination with another sparse recovery algorithm. For example, in Figure 1(c), we run SWAP using four different types of initializations. The dashed lines represent standard sparse recovery algorithms, while the solid lines with markers represent SWAP algorithms. We clearly see that all SWAP based algorithms outperform standard algorithms. Intuitively, since many sparse recovery algorithms can perform partial support recovery, using such an initialization results in a smaller search space when searching for the true support.

*Remark* 3.2. Since each iteration of SWAP necessarily produces a unique loss, the supports $S^{(1)}, \ldots, S^{(r)}$ are all unique. Thus, SWAP clearly converges in a finite number of iterations. The exact convergence rate depends on the correlations in the matrix $X$. Although we do not theoretically quantify the convergence rate, in all numerical simulations, and over a broad range of design matrices, we observed that SWAP converged in roughly $O(k)$ iterations. See Figure 1(d) for an example.

*Remark* 3.3. Using the properties of orthogonal projections, we can write Line 2 of SWAP as a difference of two rank one projection matrices. The main computational complexity is in computing

this quantity $k(p - k)$ times for all $i \in S^{(r)}$ and $i' \in (S^{(r)})^c$. If the computational complexity of computing a rank $k$ orthogonal projection is $\mathcal{I}_k$, then Line 2 can be implemented in time $O(k(\mathcal{I}_k + p - k))$. When $k \ll p$ is small, then $\mathcal{I}_k = O(k^3)$. When $k$ is large, then several computational tricks can be used to significantly reduce the computational time.

*Remark* 3.4. SWAP differs significantly from other greedy algorithms in the literature. When $k$ is known, the main distinctive feature of SWAP is that *it always maintains a $k$-sparse estimate of the support*. Note that the same is true for the computationally intractable exhaustive search algorithm [10]. Other competitive algorithms, such as forward-backwards (FoBa) [20] or CoSaMP [21], usually estimate a signal with higher sparsity level and iteratively remove variables until $k$ variables are selected. The same is true for multi-stage algorithms [22–25]. Intuitively, as we shall see in Section 4, by maintaining a support of size $k$, the performance of SWAP only depends on correlations among the columns of the matrix $X_A$, where $A$ is of size at most $2k$ and it includes the true support. In contrast, for other sparse recovery algorithms, $|A| \geq 2k$. In Figure 1, we compare SWAP to several state of the art algorithms (see Section 5 for a description of the algorithms). *In all cases,* SWAP *results in superior performance.*

## 4   Theoretical Analysis of SWAP

### 4.1   Some Important Parameters

In this Section, we collect some important parameters that determine the performance of SWAP. First, we define the restricted eigenvalue as

$$\rho_{k+\ell} := \inf \left\{ \frac{\|X\theta\|_2^2}{n\|\theta\|_2^2} : \|\theta\|_0 \leq k + \ell, |S^* \cap \text{supp}(\theta)| = k \right\}. \tag{3}$$

The parameter $\rho_{k+\ell}$ is the minimum eigenvalue of certain blocks of the matrix $X^T X / n$ of size $2k$ that includes the blocks $X_{S^*}^T X_{S^*} / n$. Smaller values of $\rho_{k+\ell}$ correspond to correlated columns in the matrix $X$. Next, we define the minimum absolute value of the non-zero entries in $\beta^*$ as

$$\beta_{\min} := \min_{i \in S^*} |\beta_i^*|. \tag{4}$$

A smaller $\beta_{\min}$ will evidently require more number of observations for exact recovery of the support. Finally, we define a parameter that characterizes the correlations between the columns of the matrix $X_{S^*}$ and the columns of the matrix $X_{(S^*)^c}$, where recall that $S^*$ is the true support of the unknown sparse vector $\beta^*$. For a set $\Omega_{k,d}$ that contains all supports of size $k$ with atleast $k - d$ active variables from $S^*$, define $\gamma_d$ as

$$\gamma_d^2 := \max_{S \in \Omega_{k,d} \setminus S^*} \min_{i \in (S^*)^c \cap S} \frac{\left\| \Sigma_{i,\bar{S}}^{S \setminus i} \left( \Sigma_{\bar{S},\bar{S}}^{S \setminus i} \right)^{-1} \right\|_1^2}{\Sigma_{i,i}^{S \setminus i}}, \bar{S} = S^* \setminus S, \tag{5}$$

where $\Sigma^B = X^T \Pi^\perp[B] X / n$. Popular sparse regression algorithms, such as the Lasso and the OMP, can perform accurate support recovery when $\zeta^2 = \max_{i \in (S^*)^c} \|\Sigma_{i,S^*} \Sigma_{S^*,S^*}^{-1}\|_1^2 < 1$. We will show in Section 3.2 that SWAP can perform accurate support recovery when $\gamma_d < 1$. Although the form of $\gamma_d$ is similar to $\zeta$, there are several key differences, which we highlight as follows:

- Since $\Omega_{k,d}$ contains all supports such that $|S^* \setminus S| \leq d$, it is clear that $\gamma_d$ is the $\ell_1$ norm of a $d \times 1$ vector, where $d \leq k$. In contrast, $\zeta$ is the $\ell_1$ norm of a $k \times 1$ vector. If indeed $\zeta < 1$, i.e., accurate support recovery is possible using the Lasso, then SWAP can be initialized by the output of the Lasso. In this case, $\gamma(\Omega) = 0$ and SWAP also outputs the true support as long as $S^*$ minimizes the loss function. We make this statement precise in Theorem 4.1. Thus, it is only when $\zeta \geq 1$ that the parameter $\gamma_d$ plays a role in the performance of SWAP.

- The parameter $\zeta$ directly computes correlations between the columns of $X$. In contrast, $\gamma_d$ computes correlations between the columns of $X$ when projected onto the null space of a matrix $X_B$, where $|B| = d - 1$.

- Notice that $\gamma_d$ is computed by taking a maximum over supports in the set $\Omega_d \setminus S^*$ and a *minimum* over inactive variables in each support. The reason that the minimum appears in $\gamma_d$ is because we choose to swap variables that result in the smallest loss. In contrast, $\zeta$ is computed by taking a *maximum* over all inactive variables.

## 4.2 Statement of Main Results

In this Section, we state the main results that characterize the performance of SWAP. Throughout this Section, we assume the following:

(A1) The observations $y$ and the measurement matrix $X$ follow the linear model in (1), where the noise is sub-Gaussian with parameter $\sigma$, and the columns of $X$ have been normalized.

(A2) SWAP is initialized with a support $S^{(1)}$ of size $k$ and $\widehat{S}$ is the output of SWAP. Since $k$ is typically unknown, a suitable value can be selected using standard model selection algorithms such as cross-validation or stability selection [26].

Our first result for SWAP is as follows.

**Theorem 4.1.** *Suppose (A1)-(A2) holds and* $|S^* \backslash S^{(1)}| \leq 1$. *If* $n > \frac{4 + \log(k^2(p-k))}{c^2 \beta_{\min}^2 \rho_{2k}/2}$, *where* $0 < c^2 \leq 1/(18\sigma^2)$, *then* $\mathbb{P}(\widehat{S} = S^*) \to 1$ *as* $(n, p, k) \to \infty$.

The proof of Theorem 4.1 can be found in the extended version of our paper [27]. Informally, Theorem 4.1 states that if the input to SWAP falsely detects at most one variable, then SWAP is high-dimensional consistent when given a sufficient number of observations $n$. The condition on $n$ is mainly enforced to guarantee that the true support $S^*$ minimizes the loss function. This condition is weaker than the sufficient conditions required for other computationally tractable sparse recovery algorithms. For example, the method FoBa is known to be superior to other methods such as the Lasso and the OMP. As shown in [20], FoBa requires that $n = \Omega(\log(p)/(\rho_{k+\ell}^3 \beta_{\min}^2))$ for high-dimensional consistent support recovery, where the choice of $\ell$, which is greater than $k$, depends on the correlations in the matrix $X$. In contrast, the condition in (4.1), which reduces to $n = \Omega(\log(p-k)/(\rho_{2k}\beta_{\min}^3))$, is weaker since $1/\rho_{k+\ell}^3 < 1/\rho_{2k}$ for $\ell > k$ and $p - k < p$. This shows that if a sparse recovery algorithm can accurately estimate the true support, then SWAP does not introduce any false positives and also outputs the true support. Furthermore, if a sparse regression algorithm falsely detects one variable, then SWAP can potentially recover the correct support. Thus, using SWAP with other algorithms does not harm the sparse recovery performance of other algorithms.

We now consider the more interesting case when SWAP is initialized by a support $S^{(1)}$ that falsely detects more than one variable. In this case, SWAP will clearly needs more than one iteration to recover the true support. Furthermore, to ensure that the true support can be recovered, we need to impose some additional assumptions on the measurement matrix $X$. The particular assumption we enforce will depend on the parameter $\gamma_k$ defined in (5). As mentioned in Section 4.1, $\gamma_k$ captures the correlations between the columns of $X_{S^*}$ and the columns of $X_{(S^*)^c}$. To simplify the statement in the next Theorem, define let $g(\delta, \rho, c) = g(\delta, \rho, c) = (\delta - 1) + 2c(\sqrt{\delta} + 1/\sqrt{\rho}) + 2c^2$.

**Theorem 4.2.** *Suppose (A1)-(A2) holds and* $|S^* \backslash S^{(1)}| > 1$. *If for a constant* $c$ *such that* $0 < c^2 < 1/(18\sigma^2)$, $g(\gamma_k, \rho_{k,1}, c\sigma) < 0$, $\log \binom{p}{k} > 4 + \log(k^2(p-k))$, *and* $n > \frac{2 \log \binom{p}{k}}{c^2 \beta_{\min}^2 \rho_{2k}^2}$, *then* $\mathbb{P}(\widehat{S} = S^*) \to 1$ *as* $(n, p, k) \to \infty$.

Theorem 4.2 says that if SWAP is initialized with *any support* of size $k$, and $\gamma_k$ satisfies the condition stated in the theorem, then SWAP will output the true support when given a sufficient number of observations. In the noiseless case, i.e., when $\sigma = 0$, the condition required for accurate support recovery reduces to $\gamma_k < 1$. The proof of Theorem 4.2, outlined in [27], relies on imposing conditions on each support of size $k$ such that that there exists a swap so that the loss can be necessarily decreased. Clearly, if such a property holds for each support, except $S^*$, then SWAP will output the true support since (i) there are only a finite number of possible supports, and (ii) each iteration of SWAP results in a different support. The dependence on $\binom{p}{k}$ in the expression for the number of observations $n$ arises from applying the union bound over all supports of size $k$.

The condition in Theorem 4.2 is independent of the initialization $S^{(1)}$. This is why the sample complexity, i.e., the number of observations $n$ required for consistent support recovery, scales as $\log \binom{p}{k}$. To reduce the sample complexity, we can impose additional conditions on the support $S^{(1)}$ that is used to initialize SWAP. Under such assumptions, assuming that $|S^* \backslash S^{(1)}| > d$, the

performance of SWAP will depend on $\gamma_d$, which is less than $\gamma_k$, and $n$ will scale as $\log \binom{p}{d}$. We refer to [27] for more details.

# 5    Numerical Simulations

In this section, we show how SWAP compares to other sparse recovery algorithms. Section 5.1 presents results for synthetic data and Section 5.2 presents results for real data.

## 5.1    Synthetic Data

To illustrate the advantages of SWAP, we use the following examples:

(A1)  We sample the rows of $X$ from a Gaussian distribution with mean zero and covariance $\Sigma$. The covariance $\Sigma$ is block-diagonal with blocks of size 10. The entries in each block $\bar{\Sigma}$ are specified as follows: $\bar{\Sigma}_{ii} = 1$ for $i \in [10]$ and $\bar{\Sigma}_{ij} = a$ for $i \neq j$. This construction of the design matrix is motivated from [18]. The true support is chosen so that each variable in the support is assigned to a different block. The non-zero entries in $\beta^*$ are chosen uniformly between 1 and 2. We let $\sigma = 1$, $p = 500$, $n = 100, 200$, $k = 20$, and $a = 0.5, 0.55, \ldots, 0.9, 0.95$.

(A2)  We sample $X$ from the same distribution as described in (A1). The only difference is that the true support is chosen so that five different blocks contain active variables and each chosen block contains *four* active variables. The rest of the parameters are also the same.

In both (A1) and (A2), as $a$ increases, the strength of correlations between the columns increases. Further, the restricted eigenvalue parameter for (A1) is greater than the restricted eigenvalue parameter of (A2).

We use the following sparse recovery algorithms to initialize SWAP: (i) Lasso, (ii) Thresholded Lasso (TLasso) [25], (iii) Forward-Backward (FoBa) [20], (iv) CoSaMP [21], (v) Marginal Regression (MaR), and (vi) Random. TLasso first applies Lasso to select a superset of the support and then selects the largest $k$ as the estimated support. In our implementation, we used Lasso to select $2k$ variables and then selected the largest $k$ variables after least-squares. This algorithm is known to have better performance that the Lasso. FoBa uses a combination of a forward and a backwards algorithm. CoSaMP is an iterative greedy algorithm. MaR selects the support by choosing the largest $k$ variables in $|X^T y|$. Finally, Random selects a random subset of size $k$. We use the notation S-TLasso to refer to the algorithm that uses TLasso as an initialization for SWAP. A similar notation follows for other algorithms.

Our results are shown in Figure 2. We use two metrics to assess the performance of SWAP. The first metric is the true positive rate (TPR), i.e., the number of active variables in the estimate divided by the total number of active variables. The second metric is the the number of iterations needed for SWAP to converge. Since all the results are over supports of size $k$, the false postive rate (FPR) is simply $1 -$ TPR. All results for SWAP based algorithms have markers, while all results for non SWAP based algorithms are represented in dashed lines.

From the TPR performance, we clearly see the advantages of using SWAP in practice. For different choices the algorithm Alg, when $n = 100$, the performance of S-Alg is always better than the performance of Alg. When the number of observations increase to $n = 200$, we observe that all SWAP based algorithms perform better than standard sparse recovery algorithms. For (A1), we have exact support recovery for SWAP when $a \leq 0.9$. For (A2), we have exact support recovery when $a < 0.8$. The reason for this difference is because of the differences in the placement of the non-zero entries.

Figures 2(a) and 2(b) shows the mean number of iterations required by SWAP based algorithms as the correlations in the matrix $X$ increase. We clearly see that the number of iterations increase with the degree of correlations. For algorithms that estimate a large fraction of the true support (TLasso, FoBa, and CoSaMP), the number of iterations is generally very small. For MaR and Random, the number of iterations is larger, but still comparable to the sparsity level of $k = 20$.

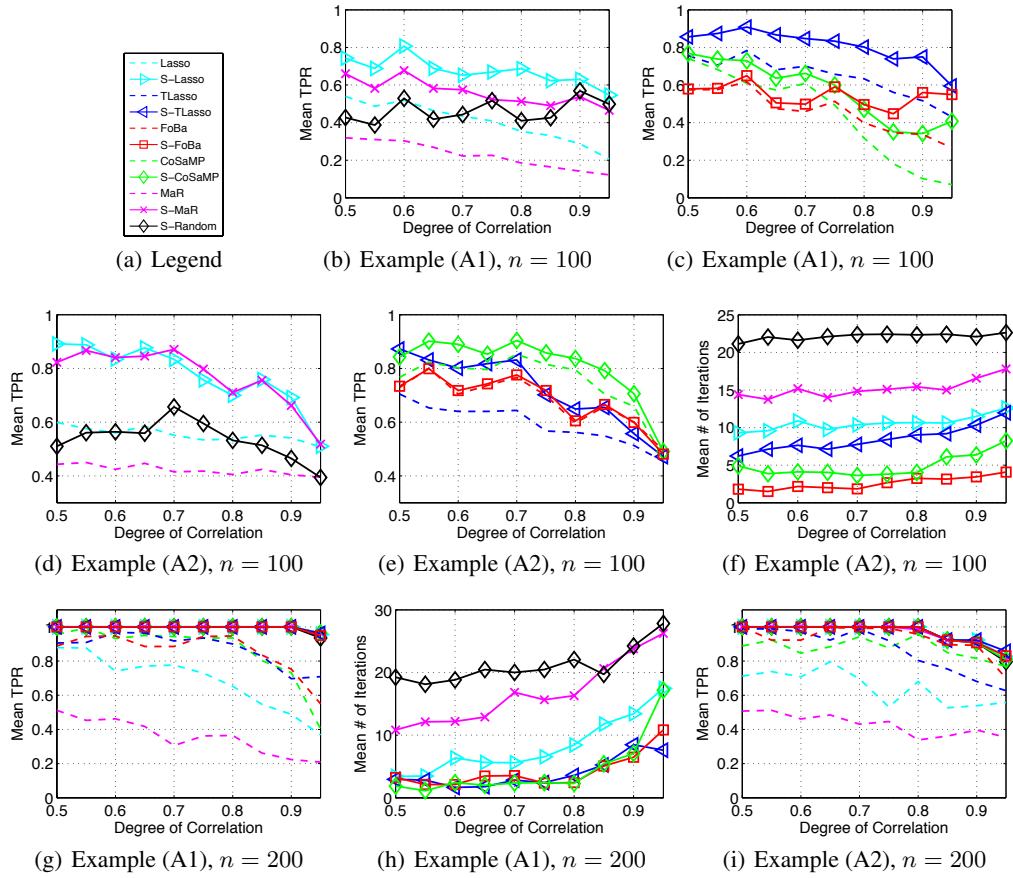

Figure 2: Empirical true positive rate (TPR) and number of iterations required by SWAP.

## 5.2 Gene Expression Data

We now present results on two gene expression cancer datasets. The first dataset[1] contains expression values from patients with two different types cancers related to leukemia. The second dataset[2] contains expression levels from patients with and without prostate cancer. The matrix $X$ contains the gene expression values and the vector $y$ is an indictor of the type of cancer a patient has. Although this is a classification problem, we treat it as a recovery problem. For the leukemia data, $p = 5147$ and $n = 72$. For the prostate cancer data, $p = 12533$ and $n = 102$. This is clearly a high-dimensional dataset, and the goal is to identify a small set of genes that are predictive of the cancer type.

Figure 3 shows the performance of standard algorithms vs. SWAP. We use leave-one-out cross-validation and apply the sparse recovery algorithms described in Section 5.1 using multiple different choices of the sparsity level. For each level of sparsity, we choose the sparse recovery algorithm (labeled as standard) and the SWAP based algorithm that results in the minimum least-squares loss over the training data. This allows us to compare the performance of using SWAP vs. not using SWAP. For both datasets, we clearly see that the training and testing error is lower for SWAP based algorithms. This means that SWAP is able to choose a subset of genes that has better predictive performance than that of standard algorithms for each level of sparsity.

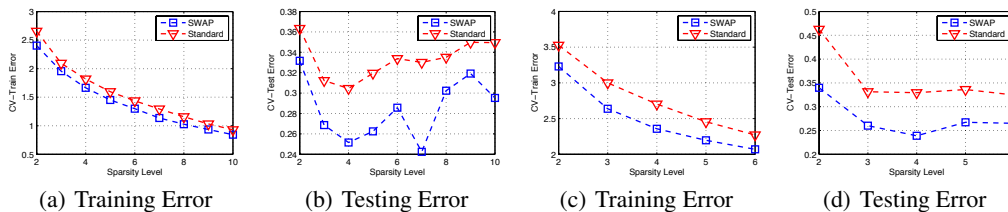

| (a) Training Error | (b) Testing Error | (c) Training Error | (d) Testing Error |

Figure 3: (a)-(b) Leukemia dataset with $p = 5147$ and $n = 72$. (c)-(d) Prostate cancer dataset with $p = 12533$ and $n = 102$.

## 6  Summary and Future Work

We studied the sparse recovery problem of estimating the support of a high-dimensional sparse vector when given a measurement matrix that contains correlated columns. We presented a simple algorithm, called SWAP, that iteratively swaps variables starting from an initial estimate of the support until an appropriate loss function can no longer be decreased further. We showed that SWAP is surprising effective in situations where the measurement matrix contains correlated columns. We theoretically quantified the conditions on the measurement matrix that guarantee accurate support recovery. Our theoretical results show that if SWAP is initialized with a support that contains some active variables, then SWAP can tolerate even higher correlations in the measurement matrix. Using numerical simulations on synthetic and real data, we showed how SWAP outperformed several sparse recovery algorithms.

Our work in this paper sets up a platform to study the following interesting extensions of SWAP. The first is a generalization of SWAP so that a group of variables can be swapped in a sequential manner. The second is a detailed analysis of SWAP when used with other sparse recovery algorithms. The third is an extension of SWAP to high-dimensional vectors that admit structured sparse representations.

## Acknowledgement

The authors would like to thank Aswin Sankaranarayanan and Christoph Studer for feedback and discussions. The work of D. Vats was partly supported by an Institute for Mathematics and Applications (IMA) Postdoctoral Fellowship.

## Footnotes

[1]see http://www.biolab.si/supp/bi-cancer/projections/info/leukemia.htm

[2]see http://www.biolab.si/supp/bi-cancer/projections/info/prostata.htm

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
