[Reviews · NeurIPS 2013]

Submitted by Assigned_Reviewer_2

Summary of the paper

This paper is concerned with the support recovery problem in linear regression in the high dimensional setup, that is to say, recovery of the non null entries in the vector of regression parameters when the number of predictors p exceeds the sample size n. A simple greedy algorithm is proposed, particularly suitable in presence of high correlation between predictors: starting from an initial guess S of the true support, it swaps each of the variables in S to each of the variables in S^c, one at a time, looking for improvement in the square loss. The sparsity level (i.e. number of zero/nonzero entries in the vector of coefficients) remains the same. Such a procedure is called SWAP, and is typically used to enhance the performances of classical sparse recovery algorithms such as the LASSO, by using the latter as the initial guess for S.

A theoretical analysis describing limitations and guarantees of SWAP are exposed in details: conditions for accurate support recovery and bounds for the number of iterations required are provided. It is shown that the required assumptions are milder than the usual irrepresentability condition. A numerical study show the benefit of using SWAP on synthetic and genomic data sets.

Comments

The paper is clearly written and pleasant to read. Good motivations and minimal tools to introduce the SWAP procedure are stated, notably due to the small example in the introductory part. Though quite simple, SWAP sounds like a good idea. A good point to my opinion is the theoretical analysis, which is nicely stated, each parameter and assumption being clearly motivated to the reader; no irrelevant or cumbersome technical points are left to the reader without explanation. This also gives insights on the limitation of the method.

Still, I a few remarks and questions:
- remark 3.3: it is not clear what "Line 3" refers to (probably line 4 in Algorithm 1?). The authors provide complexity for this famous Line 3, but the cost of the rank k projection matrix remains unclear… quantitative results would have been nice to have an idea of the additional cost over a LASSO fit, for instance. From Proposition 4.1, the number of iterations may become prohibitive as soon as there more than half of false positives in the initial guess of the support.
- with SWAP, the size of the support remains the same as the initial one: in the synthetic experiments, the authors present their results by arbitrarily fixing the support size to the true one, which may somewhat bias the results. In the genomics example, the sparsity level k varies, but obviously the best k is not always the same between SWAP and the method used for initialization. So how k is chosen practically?
- in the synthetic examples, the authors provide TPR (true positive rate) and numbers of iterations; still, the false positive rate (FPR) would have been required to judge the performance: by masking the FPR it is no clear if the methods used for the initial guess are in a regime where they fail to maintain a high TPR at a reasonable FPR.
- I am surprised by the use of classification/binary response in the genomic data considered to illustrate the performances: the square loss used to swap the variables seems inappropriate in this case; also, the methods used for building the initial guess are probably not very good at it, an enhancing the support in this situation may be easier than when considering an continuous outcome. Many genomic data set are provided with an continuous outcome.
- SWAP is appropriate to deal with correlated predictors, but none of the methods mentioned in the introduction to cope with highly correlated settings (such as the elastic-net, the Trace-Lasso, and so on) are used for comparison, although widely available.
Summary: This paper presents a simple yet appealing method with sounded theoretical justifications. It is clearly written, easy to read. Still, I have concerns regarding the numerical experiments and comparisons to state-of-the art methods. SWAP enhances some existing sparse recovery algorithms in some settings; yet it is not clear if these settings are adapted to these methods.

Submitted by Assigned_Reviewer_5

Summary:
The paper deals with a very important problem that arises in linear regression modeling when estimating high dimensional, sparse coefficient vectors (referred to as the 'sparse recovery'), namely correlations in the covariates. In this problem, one seeks a sparse set of non-zero coefficients, called the support of the coefficient vector. The true support identifies those covariates that the response variable depends on. It is well known that correlations among the covariates present a major challenge for correct identification of the support.

The authors developed a wrapper algorithm around standard sparse recovery algorithms, such as the Lasso, to improve the selected set of covariates. This wrapper, called SWAP, optimizes a desired objective function in a greedy fashion by iteratively swapping covariates. The main result of this paper is that the SWAP can recover the true set with high probability under some conditions. The authors also showed the superior performance of SWAP over the state of the art sparse recovery algorithms on synthetic and real data. There are, however, two main reasons for little satisfaction from this work:

1) Mild innovation in the actual procedure. This is foremost due to the fact that the procedure very strongly depends on initialization, i.e., on how good the initial support is in terms of its size and overlap with the true support. Given a good initial support, by the trivial swap steps the algorithm can replace the irrelevant covariates and bring in the true ones.
Thus, the authors suggest using well recognized and highly performing algorithms for sparse recovery to do the initialization. In practice, the size of the true support is not known, and to the knowledge of this reviewer there is no guarantee from the known algorithms to always recover support of correct size. This, again in practice, leads to increased run times where one has to try a (limited) range of possible support sizes. This of course leads to errors if the true support size is outside of the explored range. Also, the "quality" of the initial support is decisive of the performance, as illustrated with low true positive rates when the initial supports (of correct size) are chosen at random (figure 2).

To summarize this point, it seems the procedure itself is efficient, but seems to be designed as a very simple post-processing of the outcome of other sparse recovery algorithms, which themselves need to be quite sophisticated for the entire framework to work.

2) Mess in the results. Figure 1 has a confusing description in the text. A lot of things that are required to understand the figure are mentioned in the end of the entire discussion. Some are not formally introduced at all, like the "sparsity level" which is expressed in some numbers from 1 to 10. It would be much easier if what appears in the figure was explained right in front of the paragraphs that refer to it, and not in last remarks. Most of all, figure 2 seems to have wrong captions related to the sub-panels (a to i), which makes it really impossible to understand and to relate to conclusions made in the text. For example, in the text it is implied that Figure 2 a) and b) show numbers of iterations, where in fact a) is a legend and b) has y axis titled "Mean TPR". This makes it hard to read through and make any sense of these results.

In addition, it would be interesting to know if the "Sparse Recovery" probability is high enough in the usual settings (theorem 4.1). In particular, please report the "Sparse Recovery" probabilities for parameters that describe all the applications mentioned in the results (including the gene expression application in lines 139-146).

Finally, a comparison to other wrapper methods is missing.


Minor comments:
Line 158 With just trying several values of k and observing a linear relation between number of iterations and k, it is not possible to generalize that the run-time complexity of the SWAP is O(k).
LIne 163 please remove "roughly", the O notation is understood as "rough" already
Remark 3.2 about empirical convergence rate seems to contradict the theoretical considerations in Proposition 4.1
Line 374 "types cancers" --> "types of cancers"
Line 316 "that the Lasso" --> "than the Lasso"
Line 317 "a backwards algorithm" --> "a backward algorithms"
Line 321 "Figures 1" --> "Figure 1"
The paper is mostly nicely written and the organization of the paper is good. Howevere. the English in Section 5.1 is worse than in the rest of the paper and could be improved. Particularly, please rewrite the description for the example A1 (lines 302-308).

Summary: This work deals with an extremely important problem in high-dimensional statistics, and proposes a simple wrapper algorithm around standard sparse recovery algorithms. While this method seems to work well in practice, this paper is not novel enough (or does not bring a significant contribution) to be considered at a conference like NIPS.

Submitted by Assigned_Reviewer_6

This paper proposes a wrapper called SWAP that can be applied around a given sparse solution with the potential for improving the likelihood that the maximally sparse (L0 norm) solution can be found. The procedure involves simply optimally swapping pairs of basis vectors in the sense of minimizing a quadratic penalty term. Both theoretical and empirical evidence are provided indicating that this proposal is more robust to dictionary correlations, which may disrupt convex L1-norm-based algorithms like Lasso or greedy techniques such as OMP or CoSaMP. At a high level, the basic idea seems to be that, while most existing methods are sensitive to correlations among 2k or more dictionary columns (where k is the sparsity level of the L0 solution), the proposed SWAP algorithm is only sensitive to such correlations among 2k or fewer, a less stringent condition for guaranteeing successful sparse recovery.

One of the primary advantages of SWAP is that it can be applied to potentially improve the estimation quality of a solution produced by virtually any sparse estimation technique. Moreover, under somewhat technical conditions which depend on the cardinality and support of the maximally sparse solution, it may conceivably recovery the true support even when other algorithms fail. Although concrete examples where this provably occurs are lacking (more on this below), in general I found the paper to be mostly well-composed and thought-provoking.

A significant downside though is that the formal recovery results depend on deploying SWAP while actually knowing the cardinality k of the optimal solution, which is generally not known in practice, and it is unclear how the algorithm will degrade when k is not available in advance. This is unlike the Lasso, where provable recovery guarantees can be derived without such explicit knowledge. As a second potential limitation, while it may well be true that SWAP has not been formally proposed previously, there is likely a very practical reason for this. Namely, SWAP is not scalable to large problem sizes unlike other greedy selection methods like OMP. More concretely, in the realistic scenario when number of measurements n, basis vectors p, and the sparsity level k all increase linearly, the complexity of a single SWAP iteration would increase by at least the cube of the common scale factor. In contrast, other greedy algorithms scale no more than quadratically. Moreover, the convergence rate is still an open question. In all of the experiments provided, the sparsity k was very low, only k=20 or smaller. In such a restricted scenario, it is not surprising that a few basis functions swaps here and there can lead to considerable improvement relative to baseline, both in estimation quality (support recovery percentage) and convergence rate. It would be interesting however to see how SWAP works in a more realistic higher-dimensional setting.

The setup for Theorem 4.1, the primary result, takes a bit of time to digest and could possibly be augmented by a more comprehensive, intuitive description beyond the few remarks provided. In particular, it would be helpful to give an example dictionary such that conditions (7) and (8) are satisfied, while the corresponding conditions for Lasso, etc., are not. Note that in the experimental results section, a block diagonal dictionary is proposed; however, it would appear that with such a dictionary it is unlikely that (7) and possibly (8) would be satisfied in this case. Assuming d > 1 (meaning the initial solution given to SWAP has more than one inactive element per the definition in the paper), there will generally always be a subset S of k atoms such that all of the associated inactive elements in this subset will be correlated with an active element in the subset. This is true because any active atom has a group of highly correlated atoms by construction, most of which are not part of the active set. Consequently, unlike analogous recovery results pertaining to Lasso (which incidently do not require any knowledge of the true support pattern, only the value of k), it is much less intuitive exactly where the advantage of SWAP lies, and specific dictionary structures where Lasso provably fails and SWAP provably succeeds. In other words, it seems that the situation described on line 205 is frequently in fact true for correlated dictionaries, and hence some further comment is warranted.

Additionally, because the paper is primary of a technical nature, with involved proof techniques of the flavor developed in various other statistical settings, e.g. references [5,8–10,12], it would be helpful from a reviewing standpoint to give a few words regarding which aspects of the proof most resemble existing work and which aspects are the most novel. While I investigated parts of the proof in the lengthy supplementary section, I admittedly did not check it all (note that on line 492 of the supplementary, I believe the first instance of "active" should be switched to "inactive").


Other comments:

* I would be curious how well a slight, far more computationally tractable modification of SWAP might perform. Specifically, what if instead of searching for the best pair of atoms to swap, you instead first find the best atom to add, then find the best atom to delete, and iterate back and forth? At the very least, the complexity would decrease drastically, although it is not clear how the theoretical and empirical results might change.

* Is there a mistake on line 221 regarding the relative vector sizes? It would appear generally that max(d+1,k) = k in all of the most interesting cases (basically except when the estimated support has no overlap at all with the true support, such that d = k). Perhaps the max is supposed to be a min?

* Is there a factor of 16 missing on line 269 of the paper?

* Line 291 seems to imply that the number of measurements n need only scale logarithmically with k (and p-k). How is this possible, since doesn't this mean that n could be smaller than k at some point? Additionally, do these claims regarding the number of measurements assume that conditions (7) and (8) are somehow automatically satisfied asymptotically? Because it would seem that it would all depend on how a correlated dictionary is that is being expanded in high dimensions, which has not been formally defined in the paper unless I'm missing something obvious.

* On line 310 it does it mean that only four active variables are assigned to the same block (and the rest randomly dispersed), or alternatively that, given k = 20, five different blocks each have four active variables?
Summary: This is a solid paper presenting novel analysis of a simple algorithm, although the practical value could benefit from further empirical evidence and explanations.

Submitted by Assigned_Reviewer_7

Summary:
This paper proposes an interesting greedy algorithm called SWAP for sparse recovery from correlated measurements. The proposed algorithm iteratively swaps variables until the desired loss function cannot be decreased. The implementation is quite easy to understand and implement. The paper also presents detailed theoretical analysis that guarantees exact sparse recovery with high probability. Experimental results on both synthetic and real-world data sets show the effectiveness of the proposed algorithm.

Major Comments:
1. The paper argues that the proposed algorithm is very effective in handling measurement matrices with high correlations. The question is that why the algorithm has advantages over standard sparse learning algorithms in handling sparse problems with high correlations? Is there any intuitive explanation? How do you show the advantages of the proposed algorithm in the theoretical analysis for solving high correlated sparse learning problems?
2. Does the theoretical analysis show any advantages over other existing sparse learning algorithms with support recovery analysis? Are the assumptions weaker?
3. It seems that the assumptions in Eqs (3),(4),(7),(8) are not standard in existing sparse learning algorithms. It would be good to give intuitive explanations for these assumptions. That would be better if the authors compare the assumptions in this paper with some other commonly used assumptions.
4. Many existing sparse learning algorithms [1] [2] [3] are designed to deal with high correlated sparse learning problems. The authors should compare the proposed algorithm with those algorithms.
5. The description of the proposed algorithm in line 104 is not clear. For example, what is the definition of $L^{(1)}_{I,i^\prime}$ ?


References:
[1] Zou, H. and Hastie, T. Regularization and variable selection via the elastic net. Journal of the Royal Statistical Society. Series B, 67(2):301, 2005.
[2] Bondell, H.D. and Reich, B.J. Simultaneous regression shrinkage, variable selection and clustering of predictors with OSCAR. Biometrics, 64(1):115, 2008.
[3] Shen, X. and Huang, H.C. Grouping pursuit through a regularization solution surface. Journal of the American Statistical Association, 105(490):727–739, 2010.
Summary: This paper proposes an interesting greedy algorithm called SWAP for sparse recovery from correlated measurements. The paper also presents detailed theoretical analysis that guarantees exact sparse recovery with high probability.
Author Feedback

Author rebuttal: Reviewer 2
- The reference to Line 3 in Rem. 3.3 is a typo. Instead, it should be Line 2 (computing all the swaps)
- Prop. 4.1 is a very loose upper bound and mainly meant to show that the number of iterations is much less than the worst case bound of O(p^k).
- Our simulation results were done to complement our theoretical analysis, i.e., when presented with a support S of size k, then SWAP can find the true support under weaker conditions than all other sparse recovery algorithms. Here 'k' can be considered a regularization parameter and choosing a good 'k' is known to be a difficult problem in general. We expect to choose 'k' in practice using standard model selection algorithms such as cross-validation, BIC, EBIC, or stability selection.
- In synthetic experiments, all supports estimated have the same cardinality of 'k'. Thus, in this case, FPR = 1 - TPR. In hindsight, we should have stated this clearly in the paper for clarity.
- As mentioned in Line 65, Trace-Lasso, elastic net, etc are mainly designed to estimate a better '\beta^*'. In doing so, these algorithms estimate a superset of the true support when the correlations in the design matrix are high. In contrast, we mainly study the sparse recovery problem of estimating the true support S^*. For this reason, we did not present results on comparisons to such methods since the objective is different from the objective of the other algorithms.

Reviewer 5
Our main result is that SWAP can recover the true set under weaker conditions than that of standard sparse recovery algorithms.
- The method SWAP is indeed simple and our main contribution is in the theoretical analysis where we show that if standard sparse recovery algorithms work or miss the support by one variable, then SWAP recovers the true support under very mild conditions and with high probability. Although the SWAP algorithm is relatively straightforward, to our knowledge, prior work has not discovered the theoretical properties we outline. The extra computations by SWAP is the price to pay for having correlations in the design matrix.
- The quality of the initial support really depends on the correlations in the design matrix. If the design matrix has low correlations, then standard algorithms output "good" supports. If the design matrix has high correlations, then the output of standard algorithms will not be good and that's when SWAP is useful.
- We acknowledge that our theoretical analysis depends on k, which is like the regularization parameter for other algorithms such as the Lasso. We are not aware of any algorithms in the literature that do not depend on regularization parameters.
- We are not aware of any other wrapper methods in the literature. Any relevant references will greatly help us improve the paper.

Reviewer 6
- Here, k can be thought of as the regularization parameter. To our knowledge, provable guarantees for support recovery in all sparse recovery algorithms depend on a regularization parameter. For example, the choice of the regularization parameter in lasso depends on the noise variance (which is unknown in general).
- We acknowledge that SWAP may not be scalable. However, it is only when there exists correlations in the design matrix that we use SWAP. If the design matrix is well conditioned, then OMP, or other algorithms, will return the true support. As shown in our Theorem, if this is the case (d=0), then only one iteration of SWAP is required which has a low complexity of roughly O(k^3p). Further, our simulations show the benefits of using SWAP in practice and in all cases, the SWAP based algorithms improve the non-SWAP based algorithms.
- We mean given k = 20, five different blocks each have four active variables.
* The modification mentioned by reviewer has actually been proposed in the literature and called the forward-backward (FoBa) algorithm. The performance of this algorithm depends on restricted eigenvalues of size > 2k (see [20] in paper). Furthermore, in Fig. 1-2 we show that SWAP is able to improve FoBa's performance (see solid and dashed red lines)